# Abnormal methylation in the *NDUFA13* gene promoter of breast cancer cells breaks the cooperative DNA recognition by transcription factors

## Research Article

Transcription factor-DNA recognition; cooperative DNA binding; abnormal DNA methylation; transcription regulation; molecular dynamics simulations; bioinformatic analyses

**Author for correspondence:**
*Anna Reymer,
E-mail: anna.reymer@gu.se

Johanna Hörberg ⓘ, Björn Hallbäck ⓘ, Kevin Moreau ⓘ and Anna Reymer* ⓘ

Department of Chemistry and Molecular Biology, University of Gothenburg, Göteborg, Sweden

## Abstract

Selective DNA binding by transcription factors (TFs) is crucial for the correct regulation of DNA transcription. In healthy cells, promoters of active genes are hypomethylated. A single CpG methylation within a TF response element (RE) may change the binding preferences of the protein, thus causing the dysregulation of transcription programs. Here, we investigate a molecular mechanism driving the downregulation of the *NDUFA13* gene, due to hypermethylation, which is associated with multiple cancers. Using bioinformatic analyses of breast cancer cell line MCF7, we identify a hypermethylated region containing the binding sites of two TFs dimers, CEBPB and E2F1-DP1, located 130 b.p. from the gene transcription start site. All-atom extended MD simulations of wild type and methylated DNA alone and in complex with either one or both TFs dimers provide mechanistic insights into the cooperative asymmetric binding order of the two dimers; the CEBPB binding should occur first to facilitate the E2F1-DP1–DNA association. The CpG methylation within the E2F1-DP1 RE and the linker decrease the cooperativity effects and renders the E2F1-DP1 binding site less recognizable by the TF dimer. Taken together, the identified CpG methylation site may contribute to the downregulation of the *NDUFA13* gene.

## Introduction

DNA methylation is a common epigenetic modification, important for the regulation of neural development, neurogenesis, synaptic plasticity, brain function, immune response, and so forth (Dulac, 2010; Suarez-Alvarez *et al.,* 2012; Winnefeld and Lyko, 2012). The genome-wide DNA methylation pattern, established at an early embryonic stage, is maintained through every cell division cycle, though changes can occur as a result of environmental factors or aging (Daxinger and Whitelaw, 2010; Kaminen-Ahola *et al.,* 2010; Maegawa *et al.,* 2010) and may lead to onset of diseases, including cancers (Ehrlich, 2002; Gorelik and Richardson, 2010; Bergman and Cedar, 2013; Martincorena *et al.,* 2017). Changes in DNA methylation pattern can lead to impaired transcription of certain guard genes, and depending on the location whether in gene exons, introns, enhancers or promoters, impact transcription differently (Wang *et al.,* 2021). The molecular mechanisms for the impact of abnormal DNA methylation on transcriptional regulation vary. For example, increased methylation of CpG steps is associated with heterochromatin (Rose and Klose, 2014; Buitrago *et al.,* 2021), which makes the genome more compact and inaccessible for the transcription machinery. Alternatively, the hypermethylation in the non-coding genome may impair the binding of regulatory proteins, for example, transcription factors (TFs), or create binding sites for others (Machado *et al.,* 2015; Yimeng *et al.,* 2017).

DNA methylation is commonly found within a CpG dinucleotide, when a methyl group is added to a C5 atom of two cytosine bases on the opposite DNA strands. There are also instances of a single cytosine methylation, irrespective of the flanking nucleotide. The addition of a methyl group occurs from the major groove and does not affect the Watson–Crick base pairing, instead it modifies the conformational dynamics of DNA, increasing the overall bending and torsional rigidity (Pérez *et al.,* 2012; Lazarovici *et al.,* 2013; Carvalho *et al.,* 2014). Depending on the local sequence context, the local effects of a CpG methylation site on DNA structure can be significant (Hörberg and Reymer, 2018). The local structural changes include increased roll and decreased twist angles (Lazarovici *et al.,* 2013), and narrowing of the minor groove (Rao *et al.,* 2018). These structural changes affect DNA–protein interactions; however, the magnitude of the effect depends on the recognition motif and 3D fold of the proteins DNA-binding domains. Furthermore, the addition of a methyl group creates additional surfaces for hydrophobic interactions, which can enhance or reduce the protein–DNA binding affinities, though the effect varies depending on where in a response element (RE) the modification occurs (Kribelbauer *et al.,* 2017). Thus, understanding the changes cytosine methylation brings into solo and cooperative TFs–DNA recognition and, by extension, transcriptional regulatory processes is of great

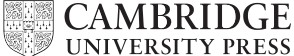

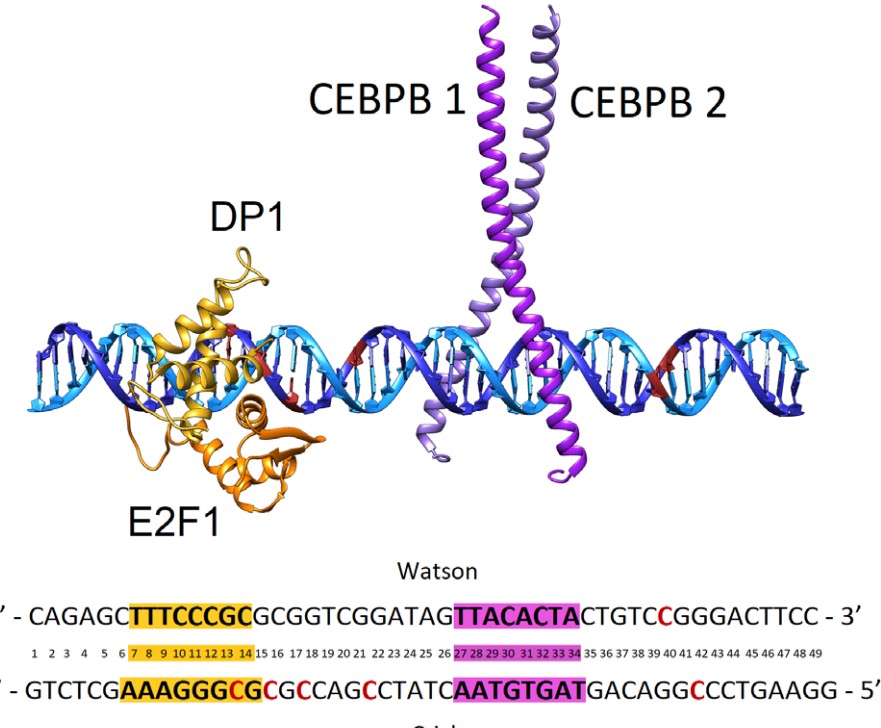

**Fig. 1.** Model system structure of the E2F1-DP1–CEBPB–DNA enhanceosome complex. DNA Watson strand (5′→3′ direction) in light blue, DNA Crick strand (3′→5′ direction) in dark blue, with cytosine residues where methylation occurs in red, E2F1 monomer in orange, DP1 monomer in yellow, CEBPB monomer 1 in magenta and CEBPB monomer 2 in purple. Below is the DNA b.p. numbering used in the model with cytosine methylation sites in red and response elements in yellow and magenta for E2F1-DP1 and CEBPB dimers, respectively.

importance for the development of early cancer diagnostics, where particular DNA methylation sites can serve as biomarkers (Levenson, 2010).

Here, we investigate a molecular mechanism driving the down-regulation of the *NDUFA13* gene, due to hypermethylation, which is linked to the onset of multiple cancers (Zhou *et al.*, 2013; Pinto and Máximo, 2018; Wang *et al.*, 2021). We employ bioinformatics analyses of breast cancer cell line MCF7 to derive our model system, based on the promoter of the *NDUFA13* gene. We identify a hypermethylated region within the promoter of the *NDUFA13* gene, 130 b.p. from the transcription start site (TSS), which contains the binding sites of two TF proteins, a homodimer CEBPB and a heterodimer E2F1-DP1 (Fig. 1). We investigate with all-atom microsecond-long molecular dynamics (MD) simulations the cooperative DNA recognition by the two TF dimers and how the process is affected by the hypermethylation. The DNA region has six methylation marks, four single methylated (ME) cytosines within the binding site of E2F1-DP1 and in the linker, and one double methylation in the flanks of the CEBPB RE. We observe that the methylation reduces the asymmetric DNA-mediated allosteric communication from the CEBPB to E2F1-DP1 binding sites, important for the tighter association of E2F1-DP1 on DNA. Furthermore, the methylation changes the physical properties of the E2F1-DP1–DNA binding site, which results in the loss of specificity in the protein–DNA contacts but has no effect on the CEBPB–DNA binding. Taken together, the identified hypermethylation marks will impede the formation of the CEBPB–E2F1-DP1–DNA enhanceosome complex, and given the close location to the TSS, may significantly contribute to the downregulation of the *NDUFA13* gene.

## Methods

### Bioinformatic analyses

The studied model system was derived from bioinformatic analyses. We employ a strategy of selecting human promoter regions where abnormal cytosine methylation is present along with TFs binding sites in the vicinity of TSSs. First, we filter out regions that possess multiple overlapping ChIP-seq signals from the ENCODE collection for breast cancer cell line MCF7 (Dunham *et al.*, 2012), retaining overlapping ChIP-seq signals within promoter regions (setting the window to 200 b.p. to TSS). Second, to further narrow the selection, the methylation pattern in promoters, obtained through reduced bisulfite seq, is analysed with UCSC Genome Browser with setting ENCODE track to 'MCF7 cells results only' (Kent *et al.*, 2002). After the filtering, the *NDUFA13* (GRIM19) promoter (Zhou *et al.*, 2013; Pinto and Máximo, 2018) is selected. Finally, we use the TFregulomeR tool (Lin *et al.*, 2020) to identify TF REs corresponding to the ChIP-seq signals near the ME CpG sites. Additionally, we analyse the availability and frequency of the identified ME CpG sites in other cancerous cells using data from http://genome.ucsc.edu/cgi-bin/hgFileUi?db=hg19&g=wgEncode HaibMethylRrbs.

### Simulated systems

We simulate eight systems, including wild type (WT) and methylated (ME) 49 b.p. long DNA molecules containing two REs of human TFs dimers: E2F1-DP1 and CEBPB, alone or in complex with either one, or both TFs dimers. The ME-systems DNA contain six ME cytosines, four single methylation sites within the E2F1-DP1

RE and the linker region, and one double methylation site in the 3′-flanking sequence of the CEBPB RE. See Fig. 1 and Supplementary Fig. S1 for WT- and ME-DNA sequences.

To design model structures of the systems, we first derive a homology model of the human E2F1-DP1 dimer with YASARA (Krieger and Vriend, 2014; Land and Humble, 2018) using as the template the crystal structure of the E2F4-DP2–DNA complex [PDB ID: 1CF7 (Zheng *et al.,* 1999)]. The sequence identity and sequence similarity (Supplementary Fig. S2*A,B*) are 63 and 71%, respectively, for the E2F1/E2F4 homologs and 89 and 95%, respectively, for the DP1/DP2 homologs, which justifies the choice of the template and ensures the quality of the model structure. Using HDOCK webserver (Yan *et al.,* 2017), we proceed with the molecular docking of two TFs dimers to B-DNA containing their corresponding REs and two flanking nucleotides on each side (XXREXX). The structure of the CEBPB dimer is adopted from the crystal structure of the CEBPB–DNA complex (PDB ID: 1GU4). We select the best TF dimer–DNA complex model based on the docking score and similarity to the corresponding crystal structures. Finally, we assemble the corresponding DNA complexes with either one TF dimer, or both in USCF Chimera (Pettersen *et al.,* 2004).

### Molecular dynamics simulations

All MD simulations were performed using the GROMACS MD engine version 2021 (Abraham *et al.,* 2015), using amber14SB (Maier *et al.,* 2015) and parmbsc1 (Ivani *et al.,* 2016) force fields for the proteins and DNA, respectively, and previously derived parameters for 5-methylcytosine (Lankaš *et al.,* 2002; Carvalho *et al.,* 2014). All simulated systems were first neutralised and then solvated in triclinic rectangular periodic boxes with SPC/E water (Mark and Nilsson, 2001) with a buffer distance of 13Å to the walls. Additional $K^+$ and $Cl^-$ ions were added to reach a physiological concentration of 150 mM KCl. Monovalent ions were treated with the Joung–Cheatham parameters (Joung and Cheatham, 2008). Using periodic boundary conditions, the systems were energy minimised with 5,000 steps of steepest decent, followed by 500 ps equilibration-runs with weak position restraints on heavy atoms of the solute (1,000 kJ/mol) in the NVT and NPT ensembles, adjusting temperature and pressure to 300 K and 1 atm (Parrinello and Rahman, 1981; Berendsen *et al.,* 1984). Releasing the restraints, for each of the eight systems, we carry out 1.1 μs MD simulations at constant pressure and temperature (1 atm and 300 K).

### Trajectory analyses

For each of the generated MD trajectories, the first 100 ns are discarded as equilibration. CPPTRAJ program from AMBERTOOLS 16 software package (Roe and Cheatham, 2013) is used for the analysis of protein–DNA contacts. We analyse both specific (hydrogen bonding and hydrophobic contacts) and nonspecific contacts (formed between either DNA or protein backbones) that are present for longer than 10% of the trajectory, see a recent publication for details (Hörberg *et al.,* 2021). Subsequently, Curves+, Canal and Canion programs (Lavery *et al.,* 2009) are used to derive the helical parameters, backbone torsional angles, groove geometry parameters and ion distributions for each trajectory snapshot extracted every ps. GROMACS energy tool is used to calculate protein–DNA interaction energies, which include short-range electrostatic and Lennard-Jones interactions. To separate interaction energies into specific and nonspecific, we calculate interactions for several atom groups that correspond to

DNA bases, protein side chains and molecule backbone. Analysis of free energies is performed using the MMPBSA/MMGBSA plugin in AMBERTOOLS 16. DNA deformation energies were derived with a multivariate Ising model (Liebl and Zacharias, 2021), which combines a harmonic deformation approximation model with an Ising model to allow for the inclusion of coupling between all conformational substates of DNA. The model has been parameterised for all tetranucleotides. The model utilises six inter-base pair (shift, slide, rise, twist, tilt and roll) and the six intra-base pair (shear, stagger, stretch, buckle, propeller twist and opening) parameters to calculate the deformation energy for a DNA sequence. Long-distance correlation analysis between DNA translational helical and groove parameters was performed in MATLAB software, according to the methods of Balaceanu *et al.* (2018). GROMACS 'covar' and 'anaeig' tools are used to derive the configurational entropies. Configurational entropies of backbone 'P' atoms of the entire DNA sequence, the E2F1-DP1-binding site (b.p. 6–15) and the CEBPB-binding site (b.p. 26–35) are derived for all systems, by removing all translational movements. From the obtained eigenvectors, the configurational entropies are calculated at 300 K using Schlitter's formula (Schlitter, 1993). To derive standard deviations, we calculate the configurational entropies for the windows 1,000 ns, last 750 ns and last 500 ns.

### Additional information

MATLAB software and Microsoft excel were used for the postprocessing and plotting of all data. USCF Chimera (Pettersen *et al.,* 2004) was used for creating all molecular graphics.

### Results

Using computational methods, we investigate a molecular mechanism for the impact of abnormal DNA methylation on transcriptional control resulting in cancers. We start with exploratory bioinformatics analyses of human genome non-coding regions, using next-generation sequencing (NGS) data. We identify in the breast cancer cell line MCF7 (Dunham *et al.,* 2012), a hypermethylated region within the promoter of the *NDUFA13* gene, 130 b.p. from the TSS (Supplementary Fig. S1). The *NDUFA13* gene encodes a NADH dehydrogenase enzyme, a part of the electron transport chain in mitochondria that can function as a tumour suppressor (Pinto and Máximo, 2018). The hypermethylation of the *NDUFA13* promoter leads to the downregulation of the gene, which increases cell proliferation and subsequently leads to the onset of breast cancer (Zhou *et al.,* 2013). The identified hypermethylated region contains in total 14 CpG ME sites (Supplementary Fig. S1*B*) within the 500 b.p. window from the TSS of the *NDUFA13* gene. Using the TFregulomeR tool (Lin *et al.,* 2020), we continue with mapping the TF REs near the hypermethylated cites that match with the ChIP-seq signals identified for the region. The analysis reveals two REs for the E2F1-DP1 heterodimer (TTTCCCGC) and the BZIP homodimer CEBPB (TTACACTA). Six out of 14 CpG methylations are located within the E2F1-DP1 RE, in the linker region between the two TF dimers, and in the 3′-flanking sites of the CEBPB RE (Fig. 1). We want to emphasise that the herein studied region with six CpG methylation sites (Chromosome 19: 19626898–19626946, see Supplementary Fig. S1) was chosen as the one with the highest potential impact on the transcriptional control, as it contains the highest density of CpG methylations and TFs binding sites. The other eight CpG

methylation sites, four further away and four closer sites to the *NDUFA13* gene-TSS may also contribute to the downregulation of the gene and deserve a separate study. Furthermore, the methylation rates (the percentage of DNA molecules that exhibit cytosine methylation at a specific CpG site) of the six CpG methylations in the MCF7 cell line vary depending on the analysing lab (Supplementary Fig. S2 and Table S1). This is not surprising as the MCF7 line is very heterogeneous (Leung *et al.*, 2014). We can speculate that the heterogeneity of the MCF7 cell line may mimic the true nature of the cancer tissue, where some cells have progressed further away with respect to the non-cancerous ones. We consider all six methylation sites to be inclusive, although they may vary from cell to cell. To address, if the identified hypermethylated region may contribute to the downregulation of the *NDUFA13* gene, we next design the E2F1-DP1–CEBPB–DNA enhanceosome complex and proceed with all-atom MD simulations.

We perform protein–DNA docking with the HDOCK server (Yan *et al.*, 2017), using the CEBPB dimer crystal structure (PDB ID: 1GU4) and a model structure of the E2F1-DP1 dimer derived through homology modelling in YASARA (Krieger and Vriend, 2014; Land and Humble, 2018), with the crystal structure of homologous E2F4-DP2 dimer [PDB ID: 1CF7 (Zheng *et al.*, 1999)] as the template. We dock each protein dimer individually to B-DNA composed of the TF RE and two adjacent flanking nucleotides (NNRENN). This yields among the top 10 highest ranked docking poses, the protein–DNA complexes resembling the expected binding modes, with RMSD of heavy atoms of 3.2 and 3.4 Å, for CEBPB– and E2F1-DP1–DNA complexes, respectively (Supplementary Fig. S4).

Then, using USCF Chimera (Pettersen *et al.*, 2004), we combine the two docked complexes through superposition on B-DNA that covers the two REs and six flanking b.p. on either 5′- and 3′-sides, 49 b.p. in total, to obtain the complete E2F1-DP1–CEBPB–DNA enhanceosome system.

We proceed with microsecond-long all-atom MD simulations for DNA alone and in complex with E2F1-DP1–CEBPB-, CEBPB-, E2F1-DP1-factors for WT and ME systems, that is, eight systems in total. Following the MD simulations, we compare the average structures of each WT system with the corresponding ME-counterpart (Fig. 2*a*). The differences are local, located within the binding region of the E2F1-DP1 dimer, where the CpG methylation sites contribute to a DNA bending towards the major groove. The differences between the WT and ME systems are more significant when the E2F1-DP1 dimer is bound to DNA (see Fig. 2*a*, Supplementary Fig. S5 and Supplementary Videos S1–S8), which is illustrated by the RMSD values of 5.6 Å for the E2F1-DP1–DNA systems and 4.5 Å for the E2F1-DP1–CEBPB–DNA systems, respectively.

### DNA deformation energies

Next, we analyse DNA deformation energy, that is the energy of DNA thermal fluctuations, in all eight MD trajectories using a multivariate Ising model, parameterised for all tetranucleotides (Liebl and Zacharias, 2021), derived to include the coupling between all possible sequence-specific conformational substates of DNA. First, we calculate the deformation energy for the region covering both TF REs and the first adjacent flanking nucleotides,

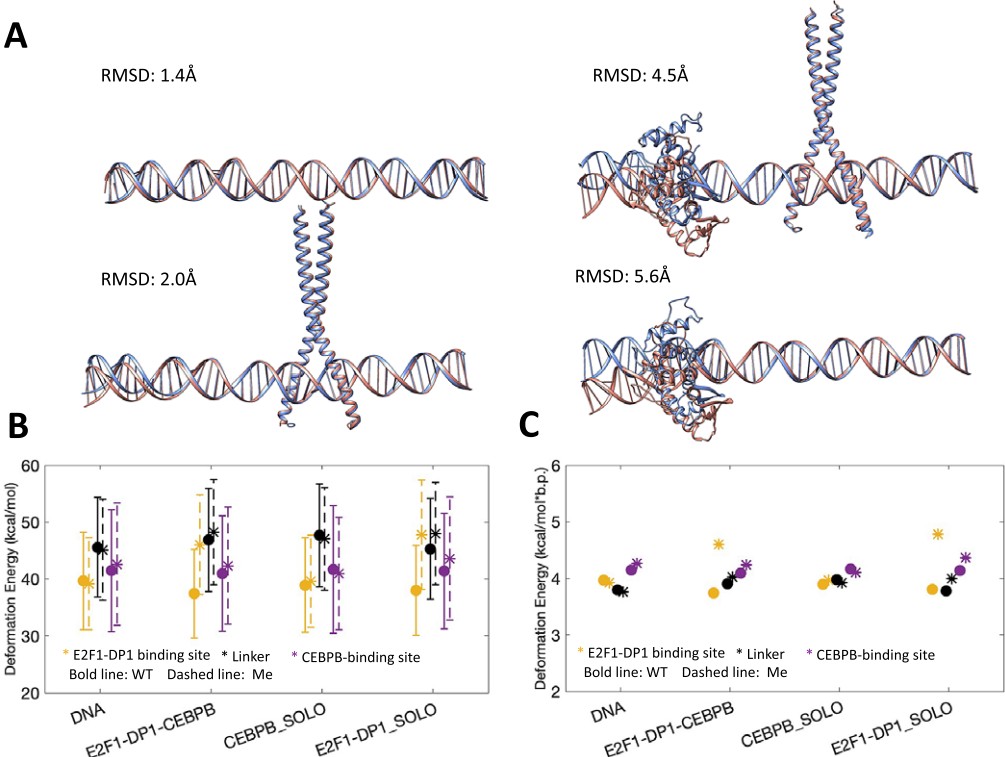

**Fig. 2.** (*a*) Comparison of average structures after 1 μs MD simulation for WT (blue) and ME (orange) systems. WT system is used as a reference, where RMSD values denote the differences in heavy atoms of DNA. (*b*) DNA deformation energies (kcal/mol) calculated with a multivariate Isling model (Liebl and Zacharias, 2021) for E2F1-DP1 binding site (b.p. 6–15), linker region (b.p. 15–26), and CEBPB binding site (b.p. 26–35). (*c*) Mean values of DNA deformation energies per b.p. (kcal/mol*b.p.) calculated for different regions of the model system, as described in panel (*b*) (with standard errors approaching zero).

**Table 1.** DNA deformation energies (kcal/mol) for the 6–35 b.p. region, calculated using a multivariate Ising model (Liebl and Zacharias, 2021)

| | DNA | E2F1-DP1–CEBPB–DNA | CEBPB–DNA | E2F1-DP1–DNA |
|---|---|---|---|---|
| WT | 120.0 ± 15.7 | 122.6 ± 15.7 | 125.5 ± 17.1 | 119.5 ± 15.6 |
| | | (2.6 / 0.09/b.p.) | (5.5 / 0.18/b.p.) | (−0.50 / −0.02/b.p.) |
| ME | 121.0 ± 15.8 | 132.3 ± 15.5 | 126.4 ± 15.4 | 134.5 ± 16.7 |
| | (1.0 / 0.03/b.p.) | (13.3 / 0.44/b.p.) | (6.4 / 0.21/b.p.) | (14.5 / 0.48/b.p.) |

*Note: Values in the parentheses correspond to the differences in the deformation energy with respect to naked WT-DNA for the entire region and per b.p.*
Abbreviations: ME, methylated; WT, wild type.

namely, 6–35 b.p. (Table 1 and Supplementary Fig. S6*A*). If a protein binding reduces the deformation energy relative to naked DNA, it indicates that the bound conformation can be readily sampled by naked DNA under physiological conditions, pointing towards a conformational selection mechanism. However, if a protein binding leads to a large increase in deformation energy relative to naked DNA, it indicates that DNA needs an external energy source to adopt the bound conformation, pointing towards an induced fit mechanism. As discussed by Battistini *et al.* (2019), a double increase or more in DNA deformation energy upon a protein binding indicates an induced fit mechanism; whereas a value between the one of naked DNA and the double increase indicates that both mechanisms co-exist. In our case, using the multivariate Isling model instead of a simple harmonic model approximation, the naked DNA thermal energy fluctuation is ~4 kcal/mol*b.p. We observe that in the WT case, DNA deformation energy slightly increases upon binding of the CEBPB dimer to naked DNA (by 0.18 kcal/mol*b.p., Table 1), while the E2F1-DP1–DNA association has no obvious impact. As stated above this is not a large increase, thus the DNA conformation for the CEBPB binding should be readily accessible within the thermal fluctuations of DNA. Upon DNA methylation, the E2F1-DP1–DNA binding further increases the deformation energy, irrespective if CEBPB is present or not (by 0.44–0.48 kcal/mol*b.p., Table 1). Although the increase is relatively low, it is a noteworthy change, which suggests that the binding of E2F1-DP1 to ME-DNA could depend on induced fit changes, which may lower the dimer–DNA affinity.

Second, we divide the DNA sequence into three regions: the E2F1-DP1 binding site, the linker, and the CEBPB binding site, and recalculate the deformation energy for each region (Fig. 2*b,c* and Supplementary Fig. S6*B–D*). This segmentation provides further insights. The deformation energy is not equally distributed over the DNA sequence. Although the standard deviations are high which make the calculated deformation energies overlap, the large sampling sets (1 million snapshots) make the standard errors approach zero. Thus, we will focus on comparing the mean values (Fig. 2*c*). In the WT case, the CEBPB–DNA binding increases the deformation energy of the linker region (by 0.17 kcal/mol*b.p.), suggesting induced conformational changes – analyses of DNA helical parameters confirm that (see below). This increase also slightly reduces the deformation energy within the E2F1-DP1 binding site (0.06 kcal/mol*b.p.), suggesting it becomes more favourable for E2F1-DP1 to bind to DNA. Nevertheless, the differences in the calculated deformation energies between all systems (Fig. 2*c* and Supplementary Fig. S6*B–D*) are relatively small, the only

noteworthy change is the increase in deformation energy of the E2F1-DP1-binding site caused by the binding E2F1-DP1 to ME-DNA (0.6–0.8 kcal/mol*b.p.). This could, as suggested, make the binding of E2F1-DP1 to ME-DNA more dependent on induced fit changes, which in turn could make the binding site less recognizable. Taken together, the analyses hint at the cooperative nature of the TF dimers–DNA association, which can be perturbed by the hypermethylation.

### Protein–DNA contacts

We continue with the analyses of protein–DNA contacts to further understand the mechanistic impact of CpG methylation on the E2F1-DP1–CEBPB–DNA enhanceosome formation. For the analyses, we employ a dynamic contacts map approach we derived earlier (Hörberg and Reymer, 2020; Hörberg *et al.*, 2021), where for each MD frame, we calculate a contact strength for every pair of protein–DNA residues that interact specifically and nonspecifically. We follow the time-evolution of the contacts strengths (Supplementary Figs S7–S10) and calculate the average strength value for each protein–DNA contact (Fig. 3 and Supplementary Figs S11, S13 and S14). The analyses of the dynamic contacts maps for the WT *versus* ME cases and solo- *versus* enhanceosome systems show that the CEBPB dimer exploits nearly the same network of contacts in all systems (Supplementary Figs S11 and S13). This suggests that the CEBPB–DNA binding is independent of CpG methylation or the binding of the E2F1-DP1 dimer. The CEBPB dimer is a BZIP factor, which recognises its REs through specific interactions formed by a conserved five residues motif (NxxAVxxSR) (Fujii *et al.*, 2000), residues 281–289, of each monomer. The studied CEBPB-RE has a non-canonical CEBPB half-site (**TAGT**GTAA), which explains the observed differences in the specific contacts exploited by the two monomers (Supplementary Figs S11 and S12). Of the five residues motif of monomer one, later CEBPB1, Ala284, Val285 and Ser288 interact hydrophobically with the flanking 5′-GpT, first TpA steps, and the adenine of the TpA step on the Crick strand (**GTA**GTGT**A**A), respectively, whereas Arg289 is involved in hydrogen bonding with the central CpA-step (TTA**CA**CTA) and the guanine of the first GpT step of the CEBPB1 half-site (TA**G**TGTAA). Though, the Arg289 contacts are somewhat weaker in the ME-systems. Specific interactions of the five residues motif of monomer two, later CEBP2, include cross-bridging hydrogen bonds by Asn281 to the first two TA b.p. (**T**TACACTA/ TAGTGT**AA**), hydrophobic contacts by Ala284, Val285 and Ser288 with the flanking 5′-GpT step and the TpT step of the CEBPB2 half-site (**GTT**ACACTA), and hydrogen bonding by Arg289 with the central TpG step (TAG**TG**TAA). Some of the abovementioned contacts exploited by Ser288 and Arg289 of both CEBPB monomers only appear in some of the simulated systems (Supplementary Figs S11 and S12). However, we believe that these contacts are not system specific, rather they are coupled to the flickering power of long-chained residues, which can fluctuate between different conformational substates.

In contrast, the DNA contacts exploited by the E2F1-DP1 dimer show a great variation among the simulated systems (Figs 3 and 4*a*, and Supplementary Fig. S14). We observe a rearrangement of the E2F1-DP1–DNA contacts when comparing the WT- and ME-E2F1-DP1–DNA systems, and a strong impact of TF-cooperativity for the E2F1-DP1–DNA binding. The E2F1-DP1 dimer belongs to the E2F-DP family of TFs, the winged-helix folded proteins that recognise their REs with a conserved four residues motif (RRxYD), residues 167–171 and 165–169 for the DP1 and E2F1 monomers,

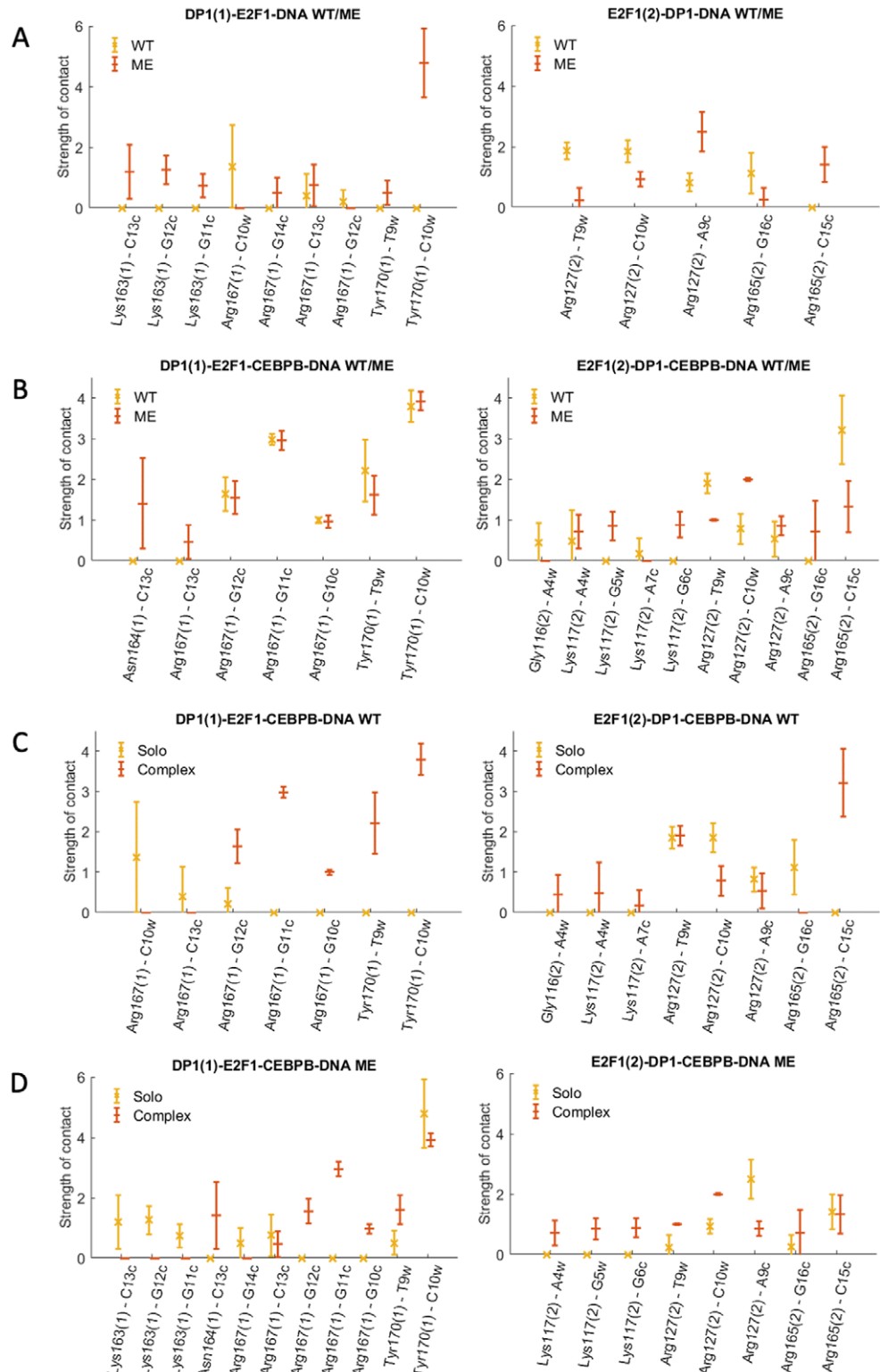

**Fig. 3.** Specific contacts between the E2F1-DP1 dimer and DNA. The plots show the strength of specific contacts formed by DP1 [marked with (1)] and E2F1 [marked with (2)] monomers in wild-type (WT) and methylated (ME) systems when bound alone to DNA (solo) (panel a) and together with CEBPB (complex) (panel b). Panels c and d show the strength of specific contacts by DP1 and E2F1 monomers in solo- versus complex-systems in WT and ME cases, respectively. For the definition of a contact strength, see Supplementary Material.

respectively (Zheng *et al.,* 1999. For DP1, Arg167 is involved in hydrogen bonding with the three guanines of DNA Crick strand (TTTCCCGC/GC**GGG**AAA) in the WT systems, in contrast the residue forms predominantly hydrophobic contacts with the methyl

group of the $C_W13$ base in the ME-systems. Tyr170 forms hydrophobic contacts with the TpC step (TT**TC**CCGC) in the WT-CEBPB-E2F1-DP1–DNA system and only with $T_W10$ base (TT**T**CCCGC) in the ME-E2F1-DP1–DNA system (Fig. 4b). For

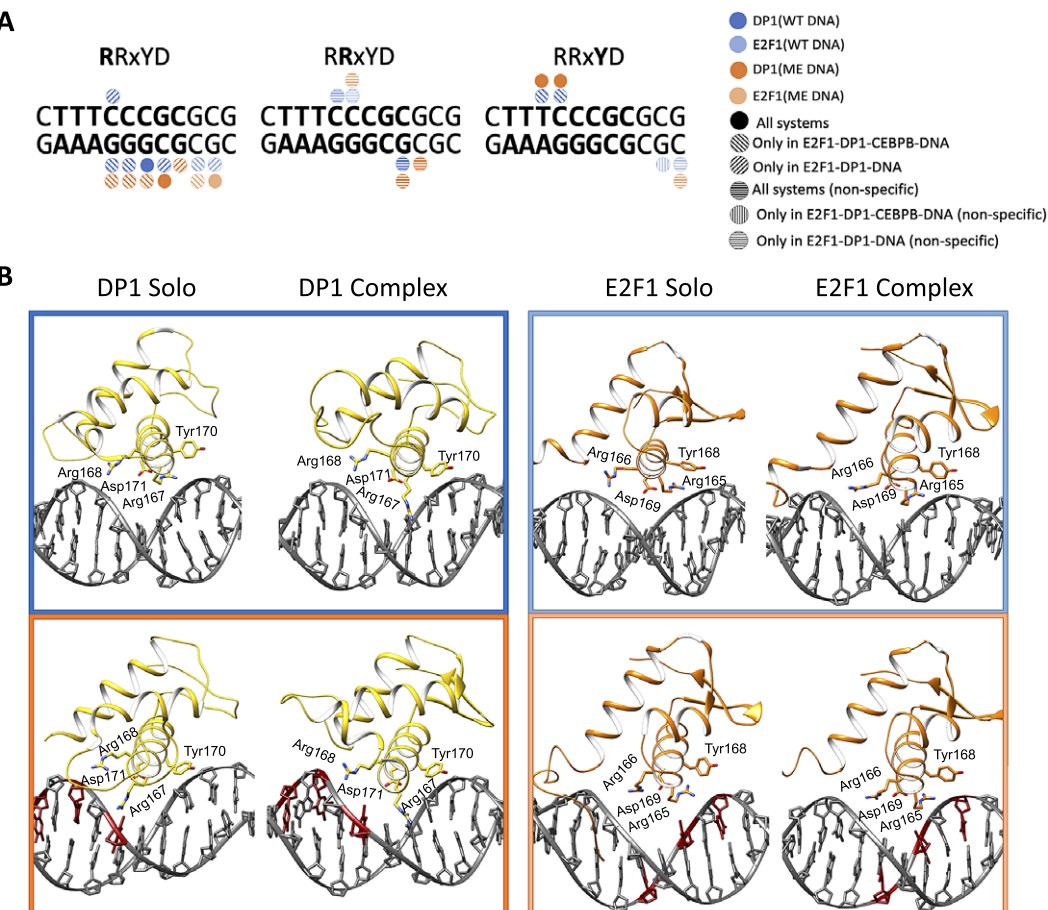

**Fig. 4.** (*a*) Schematic representation of specific protein–DNA contacts exploited by the four residues motif (RRxYD) of E2F1-DP1 in different systems. (*b*) Cartoon representation of the binding orientations and DNA contacts of the E2F1-DP1 dimer for WT (blue) and ME (orange) systems, when bound alone to DNA (solo) and together with CEBPB (complex).

E2F1, Arg165 forms hydrophobic contacts with the flanking GpC step (**GC**GCGGGAAA). For both monomers, Arg168/Arg166 interact nonspecifically with DNA backbone, while Asp171/Asp169 show no direct contact with DNA, instead stabilising the orientation of the Arg167–168/Arg165–166 residues. It should, however, be mentioned that Arg166 of E2F1 shows a reduction in the contact strength of nonspecific interactions with ME-DNA; instead, it forms mainly a salt bridge with Asp171 of DP1. This could affect the affinity of the heterodimer binding to ME-DNA. Furthermore, E2F1 has an N-terminal random-coil-tail that twists around DNA forming mostly nonspecific contacts, except for Arg127 which also specifically interacts with several b.p. of the RE (TT**TC**CCGC/CGGG**A**AA). Also, for the WT- and ME-E2F1-DP1–CEBPB–DNA systems there is an additional specific contact by Lys117 of the N-terminal tail. However, as random-coil-tails are highly flexible, exhibiting a large conformational space that is hard to sample completely, the observed variations could be simulation-specific rather than coupled to the impact of methylation.

Overall, we observe a reduction in the strength and number of specific and nonspecific DNA–protein contacts for the WT- and ME-E2F1-DP1–DNA systems (Fig. 3 and Supplementary Fig. S14). In the WT-E2F1-DP1–DNA system, the DP1 recognition helix moves out of the major groove, whereas in the ME-case, the recognition helix changes its orientation and slides downwards (Fig. 4*b*). The two binding modes of the DP1 monomer in the WT- and ME-E2F1-DP1–DNA systems lead to the loss of specificity of E2F1-DP1–DNA binding. For the WT system, the largest changes include the loss of contacts exploited by DP1 Arg167 and Tyr70, which puts the monomer in a position resembling a dissociation state. E2F1 also shows a reduction in contacts, mostly those exploited by the N-terminal tail. This could indicate that DNA is more globally flexible which makes it harder for the tail to settle. For the ME-system, DP1 shows a reduction in contacts strengths exploited by Arg167 and Tyr70; however, the effects are smaller than for the WT system. The DP1 monomer slides downwards, where Arg167 instead interacts with one of the ME cytosines, which give rise to new contacts exploited by another residue, Lys163. As for the WT system, E2F1 shows a reduction in contacts exploited by the N-terminal tail. The presence of CEBPB, secures a stable E2F1-DP1–DNA binding, with higher DNA sequence-specificity in the WT case, which implies the importance of TF-cooperativity for the E2F1-DP1–CEBPB–DNA enhanceosome formation and suggests that the CEBPB–DNA binding should occur first. Nevertheless, for the ME-systems the impact of the CEBPB presence on DNA for the E2F1-DP1–DNA association, according to the contact analysis, appears less significant indicating that the cooperativity is stronger in the WT case.

## Protein–DNA interaction energies

We proceed with the analysis of protein–DNA interaction energies. The accurate calculation of the protein–DNA interaction energy is

**Table 2.** Protein–DNA interaction energies (kcal/mol) including standard deviations, calculated between all DNA–TF-dimer complexes

|  | CEBPB–DNA | E2F1-DP1–DNA | E2F1-DP1–CEBPB–DNA | CEBPB–E2F1-DP1–DNA |
|---|---|---|---|---|
| *Specific contacts* | | | | |
| WT | −70.7 ± 11.0 (−10.5)[a] | −29.5 ± 12.2 (38.8)[b] | −60.2 ± 12.5 | −68.3 ± 13.2 |
| ME | −72.2 ± 11.6 (−12.0)[a] (2.0)[c] | −50.2 ± 10.1 (18.1)[b] (31.1)[d] | −74.2 ± 9.8 (−14.0)[a] | −81.3 ± 11.4 (−13.0)[b] |
| *Cumulative* | | | | |
| WT | −438.9 ± 63.5 (−9.0)[a] | −284.6 ± 61.8 (98.7)[b] | −429.9 ± 63.7 | −383.3 ± 51.1 |
| ME | −431.1 ± 65.1 (−1.2)[a] (13.3)[c] | −320.3 ± 60.7 (63.0)[b] (76.5)[d] | −444.4 ± 58.7 (−14.5)[a] | −396.8 ± 59.9 (−13.5)[b] |

*Note: For the E2F1-DP1–CEBPB–DNA trajectories, the provided interaction energies are for the protein dimer in bold.*
Abbreviations: ME, methylated; WT, wild type.
Difference in mean values with respect to a: WT-E2F1-DP1–**CEBPB–DNA**, b: WT-CEBPB–**E2F1-DP1–DNA**, c: ME-E2F1-DP1–**CEBPB–DNA**, d: ME-CEBPB–**E2F1-DP1–DNA**.

**Table 3.** Configurational entropies [TS (kcal/mol), for T = 300 K] for the DNA backbone P atoms

| System | Whole DNA | E2F1-DP1-binding site (b.p. 6–15) | CEBPB-binding site (b.p. 26–35) |
|---|---|---|---|
| *WT* | | | |
| DNA | 218.6 ± 0.6 | 30.2 ± 0.2 | 29.2 ± 0.3 |
| CEBPB–DNA | 207.4 ± 0.8 | 29.9 ± 0.1 | 24.3 ± 0.1 |
|  | (−11.2 ± 0.9) | (−0.3 ± 0.2) | (−5.0 ± 0.3) |
| E2F1-DP1–DNA | 207.6 ± 1.1 | 24.6 ± 0.6 | 30.3 ± 0.1 |
|  | (−11.0 ± 1.2) | (−5.7 ± 0.7) | (1.1 ± 0.3) |
| E2F1-DP1–CEBPB–DNA | 188.2 ± 0.2 | 20.3 ± 0.2 | 24.4 ± 0.6 |
|  | (−30.4 ± 0.6) | (−10.0 ± 0.3) | (−4.8 ± 0.6) |
| *ME* | | | |
| DNA | 209.3 ± 0.5 | 28.6 ± 0.2 | 29.9 ± 0.1 |
| CEBPB–DNA | 199.9 ± 0.9 | 29.6 ± 0.1 | 23.2 ± 0.3 |
|  | (−9.4 ± 1.0) | (1.0 ± 0.2) | (−6.8 ± 0.4) |
| E2F1-DP1–DNA | 201.1 ± 1.4 | 22.3 ± 1.2 | 30.1 ± 0.1 |
|  | (−8.2 ± 1.5) | (−6.3 ± 1.2) | (0.1 ± 0.1) |
| E2F1-DP1–CEBPB–DNA | 182.9 ± 1.5 | 20.2 ± 0.1 | 23.0 ± 0.5 |
|  | (−26.4 ± 1.6) | (−8.4 ± 0.22) | (−7.0 ± 0.5) |

*Note: Entropies have been derived for the windows 1,000 ns, last 750 ns, and last 500 ns. Values in parenthesis are the differences in the mean values with respect to naked DNA.*
Abbreviations: ME, methylated; WT, wild type.

difficult due to the massive negative charge of the DNA backbone. Nevertheless, for a qualitative comparison, we calculate the interaction energies including electrostatic and van der Waals, for specific and nonspecific protein–DNA contacts along all protein-bound trajectories using GROMACS energy analysis tool (Supplementary Figs S15–S20). The interaction energies for the specific and nonspecific contacts follow the trends seen in the dynamic contacts maps (Supplementary Figs S7–S10). The interaction energy distributions for the specific contacts are bimodal (Supplementary Fig. S19), which can be coupled to the flickering behaviour of the long-side chain residues. The nonspecific interaction energy distributions (Supplementary Fig. S20) are normal and broad, which reflect the large variations in the number and contact strength of most of the nonspecific contacts. When we combine the interaction energies for specific and nonspecific contacts (Table 2), it appears that cytosine methylation contributes to a stronger TF-DNA complexation. However, the differences are small with overlapping standard deviations. Furthermore, estimating interaction energies where small changes in contacts, potentially due to different conformational substates exploited by the flickering residues, can lead to overestimated changes in interaction energies. For instance, Arg289 of CEBPB1 can fluctuate between interactions with C30w, A31w and G32c, which explains most of the fluctuations in the interaction energies (Supplementary Figs S15, S16 and S19). In the WT-CEBPB–E2F1-DP1–DNA system, Arg289 shows less sampling of the Arg289-A31w state, which leads to a lower mean value of the specific interaction energy. Taken together, the analysis again indicates that TF-cooperativity greatly affects the E2F1-DP1–DNA binding. We also calculate interaction energies following the MMGBSA/MMPBSA approach (Supplementary Table S2). We exclude calculations of conformational entropies due to the size of the systems. The MMGBSA/MMPBSA energies follow the same trends, again highlighting a cooperative dependency on CEBPB for strong E2F1-DP1–DNA interactions and no deducible impact of DNA methylation.

To further address the impact of DNA methylation and TF-cooperativity, we look at configurational entropies calculated with the Schlitter's formula (Harris *et al.,* 2001) (Table 3) for the

entire DNA sequence, the E2F1-DP1-site (b.p. 6–15), and the CEBPB-binding site (b.p. 26–35) for all systems. The configurational entropy of the entire DNA sequence decreases upon the TFs binding and methylation, indicating that the molecule becomes more globally rigid. However, the rigidity is not equally distributed. Looking at the configurational entropies of the specific sites, we observe interesting mechanistic effects. First, for the WT systems, we observe that the CEBPB binding decreases the configurational entropy within the E2F1-DP1-binding site (~0.3 kcal/mol), whereas the binding of E2F1-DP1 increases the configurational entropy within the CEBPB-binding site (~1.0 kcal/mol). This indicates that CEBPB slightly reduces the global conformational flexibility of the E2F1-DP1-binding site, which may facilitate the binding of E2F1-DP1. However, the changes are small, thus, caution should be taken. Second, the configurational entropy of the CEBPB-binding site is about the same (~5 kcal/mol) when CEBPB is bound in the absence or presence of E2F1-DP1, whereas the configurational entropy of the E2F1-DP1-binding site is further reduced when E2F1-DP1 binds in the presence of CEBPB (~4 kcal/mol). This indicates that CEBPB stabilises the binding of E2F1-DP1. Third, for the ME-systems, we observe for unbound ME-DNA relative to unbound WT-DNA that the configurational entropy is reduced within the E2F1-DP1-binding site (~2 kcal/mol) and increased within the CEBPB-binding site (0.7 kcal/mol). These results indicate that DNA methylation may also have an impact on the binding of CEBPB. Fourth, the binding of CEBPB to ME-DNA increases the configural entropy of the E2F1-DP1-binding site (~1 kcal/mol) and the binding of E2F1-DP1 increases the configural entropy of the CEBPB-binding site (0.1 kcal/mol relative to ME unbound DNA and 0.8 kcal/mol relative to WT

unbound DNA). This suggests that upon methylation the order of the TF dimers binding to DNA may break.

### Changes in DNA helical and groove parameters

Next, we analyse changes in DNA helical and groove parameters for the WT *versus* ME cases and solo- *versus* enhanceosome systems (Fig. 5 and Supplementary Figs S21–S27). Previously, we reported that the binding of proteins to their REs results in small conformational adjustments in DNA helical parameters, which facilitates formation of stable specific contacts (Hörberg *et al.,* 2021). The changes are reflected in reducing the bimodality/multimodality and narrowing of the helical parameters' distributions, indicating that proteins stabilise a particular DNA substate. The analyses show that all CEBPB-bound DNA exhibit similar changes in helical and groove parameters, with respect to naked DNA, within the CEBPB RE region (b.p. 27–34). This agrees with the protein–DNA contact analyses that show nearly identical CEBPB–DNA contact networks in all systems (Supplementary Fig. S12). In addition, the CEBPB binding introduces changes in shift, slide, and grooves parameters of the four flanking b.p. on each side of the CEBPB-RE (b.p. 23–26 and 35–38), further in the linker region at $C_W20$ and in the E2F1-DP1 RE at $C_W10$ (Fig. 5). The CpG20–21 step is not involved in any interactions with the E2F1-DP1 dimer but is a soft dinucleotide step, conveniently located in the middle of the linker. The positive shift and the deeper major groove at CG10, in the presence of CEBPB, stabilise the hydrophobic contacts formed by Tyr170 of DP1, which secures the deep placement of the DP1 recognition helix into the major groove, thus explaining the nature of the observed TF-cooperativity (Fig. 4). In the ME-E2F1-DP1–CEBPB–DNA case, the effects from the CEBPB binding are smaller, due to the changes in helical shift coupled to DNA methylation (Fig. 5).

In contrast, the changes in DNA helical and groove parameters induced upon the binding of the E2F1-DP1 dimer within its RE (7–14 b.p.), the flanking regions (b.p. 3–6 and 15–18), the linker, and further in the CEBPB RE (b.p. 28, 30, 33–34) are system specific. Similarly, to the CEBPB–DNA system, the changes induced by the E2F1-DP1 dimer happen at $C_W20$. The changes in helical parameters within the CEBPB RE, induced by the E2F1-DP1 binding, do not significantly alter the CEBPB–DNA specific contacts network, as mentioned above. Methylation of CpG steps leads to a decreased twist and an increased roll (Supplementary Figs S23 and S24), which through DNA backbone BI → BII transitions, induce changes in shift and slide, and consequently in major groove depth and width (Fig. 5 and Supplementary Figs S21, S22, S26 and S27). Focusing on shift, where we observe the opposite sign of shift between the WT and ME cases for the E2F1-DP1-bound systems for the last three b.p. steps of E2F1-DP RE (TTTCC**CGCG**). The effect is more obvious when we analyse the b.p. displacement in the *x*-axis (Supplementary Fig. S25). This explains the rearrangement of the DNA contacts formed by the E2F1-DP1 dimer, and the appearance of the new specific contacts by Lys163 and Asn164 of DP1 for the ME-E2F1-DP1–DNA and ME-E2F1-DP1–CEBPB–DNA systems, respectively (Fig. 4). For the linker region, in addition to the change of the shift sign for the ME b.p., we also observe the narrowing of the shift distributions (Supplementary Fig. S21B). This indicates that DNA methylation stabilises different from the WT case conformational substates, which explains the damping of the allosteric communication between the two TF dimers.

Previous studies have shown that cooperativity between two TFs, which do not exhibit direct protein–protein interactions, arises due to DNA-mediated allosteric signals transmitted by one TF to the binding site of the other TF in a time-dependent manner (Balaceanu *et al.,* 2018). Thus, we next analyse long-distance correlations of DNA groove parameters and helical translational parameters with and without a time lag for the entire trajectories (0.1–1.1 μs) (Fig. 6 and Supplementary Figs S28 and S29). Our results show, in agreement with the previous studies, that the binding of CEBPB, but not E2F1-DP1, results in long-distance correlations of DNA b.ps. parameters (Balaceanu *et al.,* 2018). The strongest impact of the CEBPB binding on long-distance correlations is seen for the shift parameter and the major groove width (Fig. 6). The signal is translated from $C_W32$ in the middle of the CEBPB1 RE, over to $G_W21G_W22A_W23$ and then from $T_W19C_W20$ in the linker region to the E2F1-DP1 RE b.p. 10–14 (TTT**CCCGC**). However, the correlation coefficients are small (<0.3), therefore caution should be taken. The addition of a time lag has not improved the long-distance correlations. Furthermore, the long-distance communication between the two binding sites increases when both TFs are bound to DNA. This could indicate that cooperativity and communication first arise when one TF is bound to DNA and the other slides and approaches its binding site. The long-distance correlations somewhat increase (<0.35) when calculated for the trajectories intervals when the flickering residues form specific contacts with DNA. However, we believe that taking the complete trajectories provides a more complete picture of the process. The described long-distance correlations observed within the linker and E2F1-DP1 RE regions upon the CEBPB binding are missing when DNA becomes ME (Supplementary Figs S28 and S29), suggesting that methylation hinders the propagation of the allosteric signal.

### Ion populations

Previous studies have shown that counterion populations within DNA grooves are sequence-specific: depending both on sequence-specific flexibility of the DNA backbone and electronegative groups in DNA bases, and may impact the TF–DNA association (Hud and Polak, 2001; Dans *et al.,* 2014; Pasi *et al.,* 2015). Thus, we also investigate the counterion populations within the major and minor grooves for the naked and protein-bound WT- and ME-DNA systems, using Canion program. First, for the WT systems, we observe an elevated K+ concentration at C10 and C14 positions in the absence of CEBPB (Supplementary Fig. S30*A*). Second, we observe a slight increase in K+ molarity in the major groove of the E2F1-DP1 RE region for naked and CEBPB-bound ME-DNA, in comparison to the corresponding WT systems. In addition, the analyses show that cytosine methylation affects the radial distribution of K+ ions within the major groove; the ions are shifted by 0.5–1.0 Å closer to DNA helical axis with respect to WT-DNA (Supplementary Fig. S28*B*). These observations agree with the conclusions derived through dynamic contacts maps showing that the recognition helix of DP1 moves out of the major groove in the WT-E1F2-DP1–DNA case; and in the ME-E1F2-DP1–DNA case, the helix slides downwards reducing the specificity of the DP1–DNA binding.

### Discussion

Using NGS data on breast cancer cell line MCF7, we have investigated the mechanistic impact of abnormal DNA methylation in the non-coding genome on the downregulation of the *NDUFA13* gene. The downregulation of the *NDUFA13* gene, which encodes a

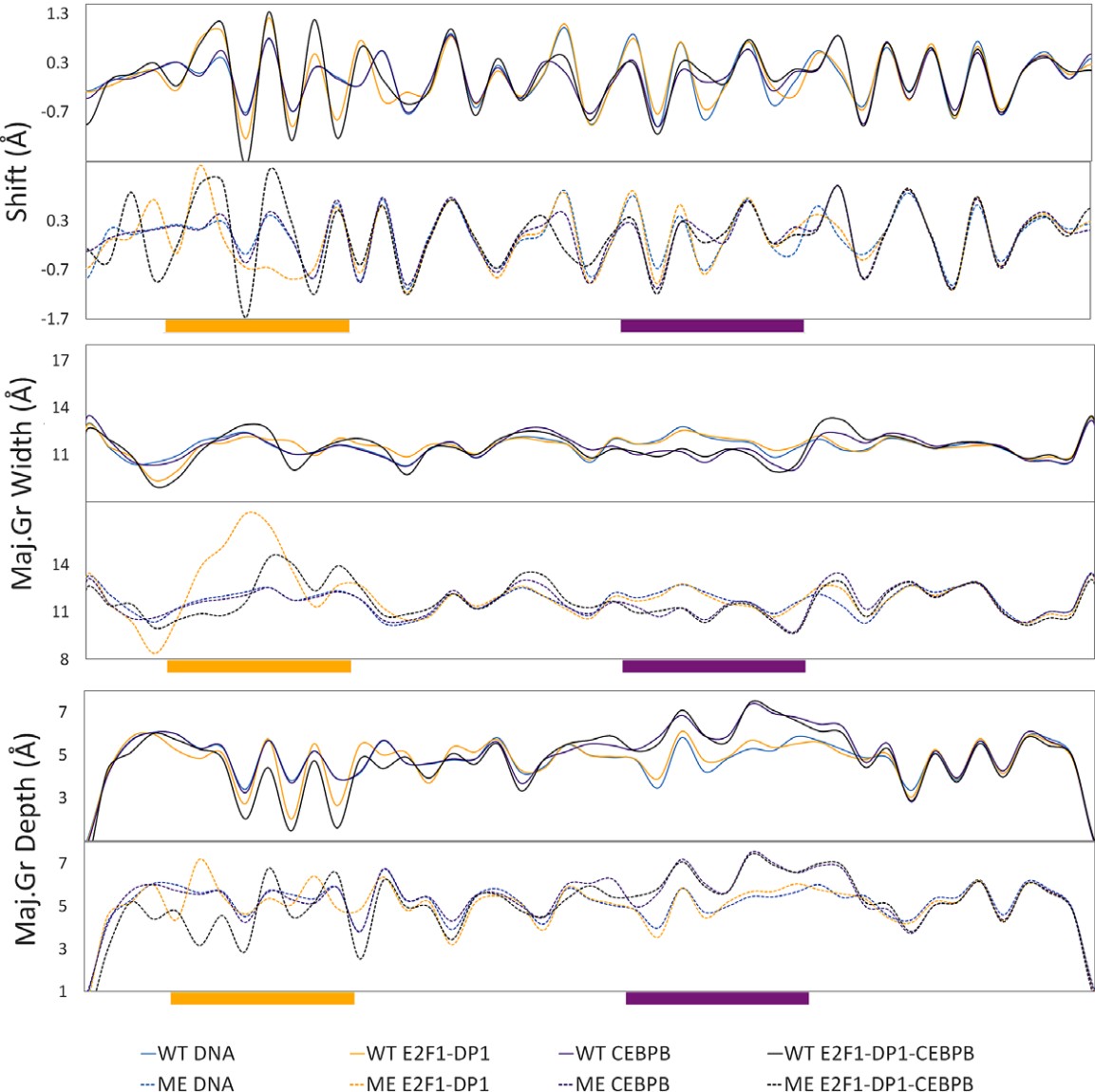

**Fig. 5.** Average values for shift, major groove width and depth (in Å) for wild-type (WT) and methylated (ME) DNA alone and in complex with either CEBPB/E2F1-DP1 dimers, or the complete CEBPB–E2F1-DP1 enhanceosome. The WT values are depicted with bold lines, ME – with dashed lines.

tumour suppressor NADH dehydrogenase enzyme, is observed in multiple cancers (Zhou *et al.,* 2013; Pinto and Máximo, 2018; Wang *et al.,* 2021). We have identified a hypermethylated DNA region 130 b.p. from the *NDUFA13* gene TSS, which also contains the DNA binding sites for two TFs dimers, CEBPB and E2F1-DP1. The proximity to the TSS and overlap with the TFs REs suggest that this hypermethylated site may contribute to the downregulation of the *NDUFA13* gene. Furthermore, our bioinformatic analysis confirms same CpG methylation sites are also present, with various methylation rates, in other cancer cell lines (Supplementary Fig. S2 and Table S1). To verify the hypothesis, we have performed all-atom microsecond-long MD simulation of WT- and ME-DNA alone and in complex with either solo CEPBP/E2F1-DP1 dimers, or E2F1-DP1–CEBPB–DNA enhanceosome systems. We analysed protein–DNA interactions, changes in DNA conformational dynamics, and ion distributions. Our data allow to describe with atom-level detail the mechanism of cooperative DNA recognition by the two TF dimers, and how the process is affected by DNA hypermethylation.

In terms of cooperative DNA recognition, we see that the primary binding of CEBPB stabilises the E2F1-DP1–DNA association. This is reflected by the stronger E2F1-DP1–DNA interaction energies and by the binding orientation of the E2F1-DP1 dimer, deeper within DNA major groove, in the presence of CEBPB. Through alterations in DNA helical and groove parameters in the linker region, the primary CEBPB–DNA binding induces local structural changes within the E2F1-DP1 RE that facilitate the formation of the specific contacts by Tyr170 of DP1. Our data suggest that Tyr170–DNA specific contacts anchor the E2F1-DP1 dimer into DNA major groove. This agrees well with a recently published statistical analysis of SELEX data that identifies DNA shape as a major component of TF-cooperativity (Ibarra *et al.,* 2020).

Interestingly, the primary E2F1-DP1–DNA binding also induces changes in DNA helical and groove parameters within the CEBPB RE, however, they do not noticeably affect the CEBPB–DNA contact network. This agrees with our previous study

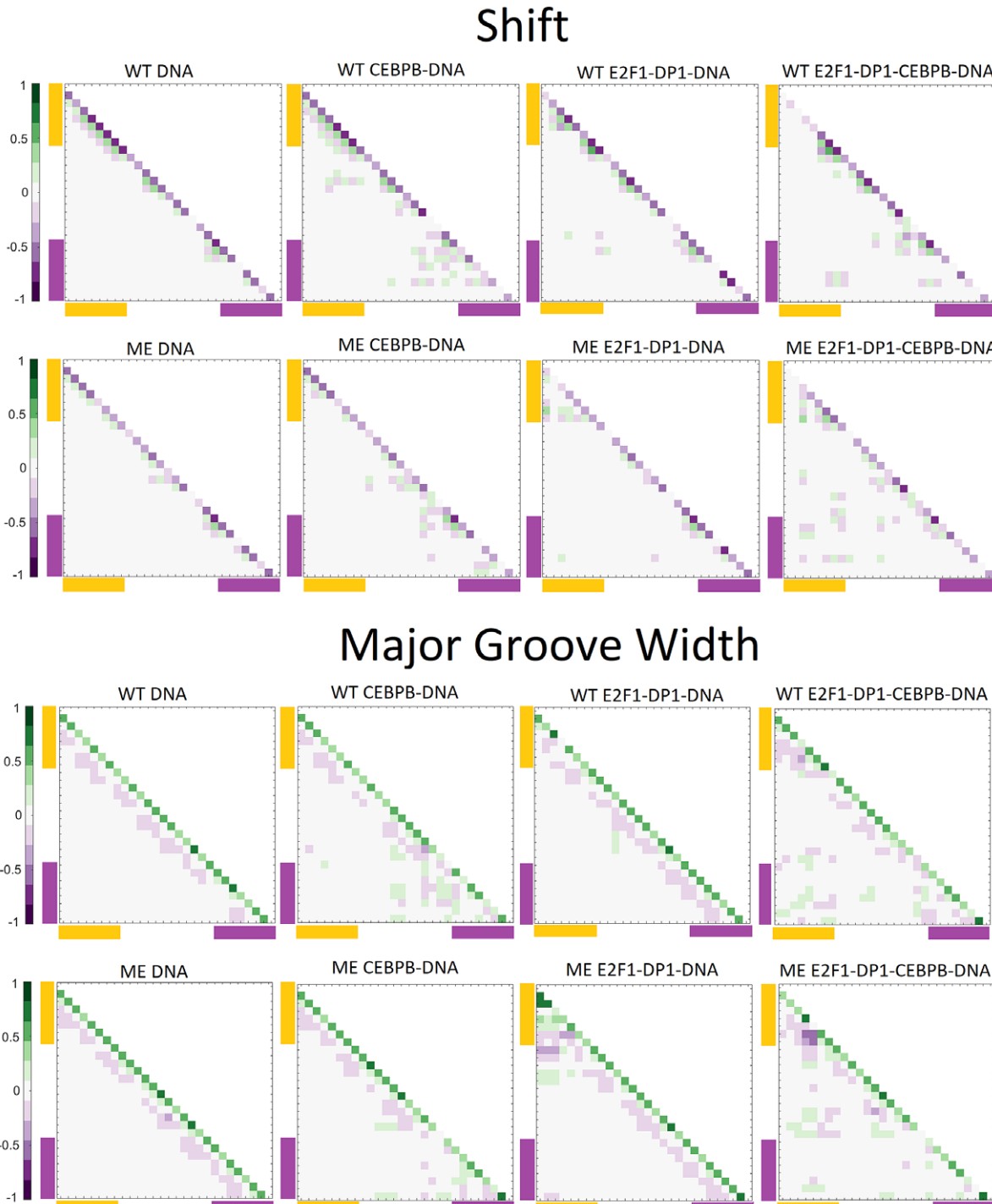

**Fig. 6.** Correlation coefficient maps without time lag for b.p. shift and major groove width along the WT and ME sequences for DNA alone and in complex with either CEBPB/E2F1-DP1 dimers, or with both TFs dimers CEBPB–E2F1-DP1. Correlation coefficients are calculated for the entire (0.1–1.1 μs) trajectories. In all panels, the E2F1-DP1 response element is marked with yellow and the CEBPB response element is marked with magenta.

that shows that BZIP TFs possess a great adaptability to changes in DNA structure (Hörberg and Reymer, 2020). This, in turn, allows us to hypothesise that CEBPB–DNA binding will happen first, which will facilitate the E2F1-DP1–CEBPB–DNA enhanceosome formation. However, the strongest DNA-mediated communication between the TFs dimers, through increased number of long-distance correlations of helical and groove parameters, arises only when both TFs dimers are bound to DNA. This is different from the

previously published mechanism for the BAMHI-GRDBD cooperativity (Balaceanu *et al.,* 2018), which arises due to time-dependent allosteric signals transmitted by BAMHI to the binding site of GRDBD. This illustrates that there is no universal mechanism of TF-cooperativity. In the case of E2F1-DP1-CEBPB cooperativity, the DNA-mediated allosteric communication between the two binding sites might first be sensed when CEBPB is bound to DNA and E2D1-DP1 is approaching its RE.

In terms of the CpG methylation impact, E2F1-DP1 has been reported to be deficient in binding to ME-DNA (Campanero *et al.*, 2000). However, the ability of E2F1-DP1 to bind DNA may depend on DNA sequence-specific flexibility and whether the binding site contains a single or a double CpG methylation. The here-studied DNA sequence contains single ME CpG marks within the E2F1-DP1-RE and linker regions, and a double ME CpG in the 3′-flanks of the CEBPB-RE. Our data show that CpG methylation within the E2F1-DP1-RE leads to different conformational substates in DNA helical and groove parameters, making DNA wider and shallower in the major groove, which results in the loss of specificity for some E2F1-DP1–DNA contacts, but overall, a stronger E2F1-DP1–DNA association. Yet, our analyses reveal that the DNA deformation energy increases in the ME-E2F1-DP1–DNA system, in comparison with the WT case. This suggests that the E2F1-DP1 association with ME-DNA will cost more energy, which may imply that the E2F1-DP1 will no longer recognise it as a true binding site (Battistini *et al.*, 2019). Furthermore, CpG methylation affects the structural changes within the E2F1-DP1 RE induced by the CEBPB–DNA binding, which is reflected, for example, by the loss of bimodality in shift and groove parameters in the linker region. We also observe that the long-distance correlations of DNA helical and groove parameters, seen for the WT case, disappear. Finally, the double CpG methylation in the 3′-flanks of the CEBPB RE seems to have no effect on either the CEBPB- or E2F1-DP1–DNA binding. Taken together, our data suggest that abnormal methylation in the noncoding genome may impede the cooperative DNA recognition by collaborative TF proteins, which could impact gene transcription of the NUDFA13 gene and contribute to its downregulation. We see our results as a starting point for further experimental validation that will or will not show that the ability of CEBPB and E2F1-DP1 TF dimers to cooperatively bind to DNA upon methylation affects the *NDUFA13* gene transcription and, potentially, lead to the onset of cancers. If so, this will open new avenues for cancer diagnostics.

**Open peer review.** To view the open peer review materials for this article, please visit http://doi.org/10.1017/qrd.2022.21.

**Supplementary Materials.** To view supplementary material for this article, please visit http://doi.org/10.1017/qrd.2022.21.

**Data availability statement.** All data generated and analysed in this study are available from the corresponding author upon request.

**Acknowledgments.** The authors thank Swedish National Infrastructure for Computing (SNIC) for the generous provision of computing resources.

**Author contributions.** All authors conceived and designed the study. J.H. and B.H. performed the MD simulations. K.M and A.R. performed the bioinformatics analyses. J.H., B.H. and A.R. analysed the data. J.H. and A.R. wrote the article.

**Financial support.** Swedish Foundation for Strategic Research SSF Grant [ITM170431] and Magn. Bergvalls Foundation Grant to A.R.; Lawski Foundation Stipend to B.H.

**Conflict of interest.** The authors declare no conflicts of interest.

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
