## [Reviewer Report]

*Comments to Author*: In this publication, the authors provide all-atom modelling description on the effect of methylation to the cooperative binding of two transcription factors at the promoter of the oncogene NDUFA13. The working hypothesis is that this hypermethylated site reduces the affinity to the transcription factors and, as a consequence, downregulates the gene. Thus, this study follows an interesting and ambitious approach of connecting all-atom detail with biological information that can have clinical applications, as this methylated side can be used as biomarker. The study is also done with care and explains well how DNA methylation reduces the cooperative effect. However, the authors find interaction energies to be more favourable to methylated DNA than to ‘wild’ DNA, which go against their hypothesis. Given the length of their simulations, wouldn’t be worth to calculate configurational entropies and their extrapolation to infinite time (https://doi.org/10.1021/ja016233n)? They have been detected key in cooperativity (for example in https://doi.org/10.1021/ja016233n). Without clarifying this, the impact of the paper substantially diminish, which I think is a shame.

Maybe the presented ‘deformation energies’ could be helpful in this sense?Although I find them very confusing as they are described in the manuscript. I understanddeformation energy as the needed energy to deform DNA from relaxed state to a target state. If so, which is the target state? At some point the concept of ‘bioactive conformation’ is mentioned, but it is no explain what it is. Also, why naked DNA, which is relaxed, has an energy different than zero? What a ‘DNA deformation energy increase’ mean?

In my opinion, the relevance of their study could also be increased by giving more context on their bioinfomatics analysis and adding discussion on these terms. Is the chosen sequence hypermethylated in other cancer cell lines? In that case, the prediction of this site as a biomarker could be better established. Is there any reason to choose MCF7 cell line? Also, the authors did a genome-wide search for detecting cases with pairs of transcription factors affected by methylation. Are there many? How general this found effect might be?

Other comments

Page 7. Following with deformation energies, it is not clear which criteria use the authors to establish where there is a change and where there is not. To me, all distributions seem well overlap. I think it is fair for the authors to talk about trends, but as it is written, they seem to claim significant changes, which is not clear to me.

Page 7. DNA deformation energies for the different parts. I really do not appreciate the difference here that protein binding brings to the linking zone. Maybe it is due to the inherent variability of the energies? Would plotting standard errors instead of deviation help here?

End of page 9. In the protein-DNA contacts there is a clear difference between the different simulations. However it is not directly explain in the text which are the main differences. The sentence “Overall, we observe a reduction in the strength and the number of specific and nonspecific DNA-protein contacts for WT- and ME-E2F1-DP1-DNA systems (Figures 3 and S12).” do not specify which is the one with reduced interactions with respect the other. In addition, at the end of this paragraph, I think it would be useful to explain which is the specific effect of methylation in the presence of CEBPB-DNA

Table 1 and 2. I think plotting difference energies with respect to DNA or similar (making them as reference so putting them as zero), would help to appreciate the magnitude of cooperativity vs the effect of methylation, and would help to discuss the balance between these two effects.

Protein-DNA interaction energies. I have several comments here:

1) “The interaction energies for the specific and nonspecific contacts follow the trends seen in the dynamic contact maps (Figures S5-S8)”. Which are these trends? I can’t see them looking the figures. This might help to understand the result of interaction energies.

2) “However, the differences are small with overlapping standard deviations, which can be coupled to the different conformational substates exploited by the flickering residues.” Can we understand better these substates?

3) The authors should be consistent with the criteria of using overlap in standard deviations to state a difference as in Table 2 and Figure 2.

4) “Also, we should remember about the DNA deformation energies, which significantly increase when the methylation marks appear within the E2F1-DP1 binding site.” Is the effect on DNA deformation bigger than on affinity? Which is the resultant balance?

Minor comments

For the clarity of the reader, I would refer to Figure 1 in the last paragraph of Introduction.

Which parameters have been used to describe ions?

RE is defined in Result section, but it should be defined the first time it appears in the text, which is in Method section.

Figure 2. In A, it is not clear which structure is being used as a reference to calculate RMSd.

Lines in B and C are so thin they are difficult to see.

Page 8, line 19: “aar”

Figure 4 is very nice, but I wonder if it would help the reader to mark with lines when the interaction is form between the different aa and the DNA. In addition, I wonder if it would be informative to include an image of the global changes on the enhanceosome

Figure 5. Lines are too thin again.

The statement “Supplementary Data are available at NAR online” should be updated

---

## [Reviewer Report]

*Comments to Author*: Reviewer #1: In this publication, the authors provide all-atom modelling description on the effect of methylation to the cooperative binding of two transcription factors at the promoter of the oncogene NDUFA13. The working hypothesis is that this hypermethylated site reduces the affinity to the transcription factors and, as a consequence, downregulates the gene. Thus, this study follows an interesting and ambitious approach of connecting all-atom detail with biological information that can have clinical applications, as this methylated side can be used as biomarker. The study is also done with care and explains well how DNA methylation reduces the cooperative effect. However, the authors find interaction energies to be more favourable to methylated DNA than to ‘wild’ DNA, which go against their hypothesis. Given the length of their simulations, wouldn’t be worth to calculate configurational entropies and their extrapolation to infinite time (https://doi.org/10.1021/ja016233n)? They have been detected key in cooperativity (for example in https://doi.org/10.1021/ja016233n). Without clarifying this, the impact of the paper substantially diminish, which I think is a shame.

Maybe the presented ‘deformation energies’ could be helpful in this sense?Although I find them very confusing as they are described in the manuscript. I understanddeformation energy as the needed energy to deform DNA from relaxed state to a target state. If so, which is the target state? At some point the concept of ‘bioactive conformation’ is mentioned, but it is no explain what it is. Also, why naked DNA, which is relaxed, has an energy different than zero? What a ‘DNA deformation energy increase’ mean?

In my opinion, the relevance of their study could also be increased by giving more context on their bioinfomatics analysis and adding discussion on these terms. Is the chosen sequence hypermethylated in other cancer cell lines? In that case, the prediction of this site as a biomarker could be better established. Is there any reason to choose MCF7 cell line? Also, the authors did a genome-wide search for detecting cases with pairs of transcription factors affected by methylation. Are there many? How general this found effect might be?

Other comments

Page 7. Following with deformation energies, it is not clear which criteria use the authors to establish where there is a change and where there is not. To me, all distributions seem well overlap. I think it is fair for the authors to talk about trends, but as it is written, they seem to claim significant changes, which is not clear to me.

Page 7. DNA deformation energies for the different parts. I really do not appreciate the difference here that protein binding brings to the linking zone. Maybe it is due to the inherent variability of the energies? Would plotting standard errors instead of deviation help here?

End of page 9. In the protein-DNA contacts there is a clear difference between the different simulations. However it is not directly explain in the text which are the main differences. The sentence “Overall, we observe a reduction in the strength and the number of specific and nonspecific DNA-protein contacts for WT- and ME-E2F1-DP1-DNA systems (Figures 3 and S12).” do not specify which is the one with reduced interactions with respect the other. In addition, at the end of this paragraph, I think it would be useful to explain which is the specific effect of methylation in the presence of CEBPB-DNA

Table 1 and 2. I think plotting difference energies with respect to DNA or similar (making them as reference so putting them as zero), would help to appreciate the magnitude of cooperativity vs the effect of methylation, and would help to discuss the balance between these two effects.

Protein-DNA interaction energies. I have several comments here:

1) “The interaction energies for the specific and nonspecific contacts follow the trends seen in the dynamic contact maps (Figures S5-S8)”. Which are these trends? I can’t see them looking the figures. This might help to understand the result of interaction energies.

2) “However, the differences are small with overlapping standard deviations, which can be coupled to the different conformational substates exploited by the flickering residues.” Can we understand better these substates?

3) The authors should be consistent with the criteria of using overlap in standard deviations to state a difference as in Table 2 and Figure 2.

4) “Also, we should remember about the DNA deformation energies, which significantly increase when the methylation marks appear within the E2F1-DP1 binding site.” Is the effect on DNA deformation bigger than on affinity? Which is the resultant balance?

Minor comments

For the clarity of the reader, I would refer to Figure 1 in the last paragraph of Introduction.

Which parameters have been used to describe ions?

RE is defined in Result section, but it should be defined the first time it appears in the text, which is in Method section.

Figure 2. In A, it is not clear which structure is being used as a reference to calculate RMSd.

Lines in B and C are so thin they are difficult to see.

Page 8, line 19: “aar”

Figure 4 is very nice, but I wonder if it would help the reader to mark with lines when the interaction is form between the different aa and the DNA. In addition, I wonder if it would be informative to include an image of the global changes on the enhanceosome

Figure 5. Lines are too thin again.

The statement “Supplementary Data are available at NAR online” should be updated

---

## [Reviewer Report]

*Comments to Author*: The authors have satisfactorily addressed the referees comments. Before final submission, I would ask them to very carefully proof-read their corrected manuscript for typographical and grammatical errors. For example, on line 30 below Table 2, “causation” is not the correct word - caution would be better. When referencing Battistini et al, the authors probably didn’t want to refer to Modesto and co-workers.